# ChatGPT Asks, BLIP-2 Answers: Automatic Questioning Towards Enriched Visual Descriptions

**Deyao Zhu, Jun Chen**[*]**, Kilichbek Haydarov**[*]**,**
**Xiaoqian Shen, Wenxuan Zhang, Mohamed Elhoseiny**
*King Abdullah University of Science and Technology*
*{deyao.zhu, jun.chen, kilichbek.haydarov, xiaoqian.shen,*
*wenxuan.zhang, mohamed.elhoseiny}@kaust.edu.sa*
**Reviewed on OpenReview:** *https://openreview.net/forum?id=1LoVwFkZNo*

## Abstract

Asking insightful questions is crucial for acquiring knowledge and expanding our understanding of the world. However, the importance of questioning has been largely overlooked in AI research, where models have been primarily developed to answer questions. With the recent advancements of large language models (LLMs) like ChatGPT, we discover their capability to ask high-quality questions when provided with a suitable prompt. This discovery presents a new opportunity to develop an automatic questioning system. In this paper, we introduce ChatCaptioner, a novel automatic-questioning method deployed in image captioning. Here, ChatGPT is prompted to ask a series of informative questions about images to BLIP-2, a strong vision question-answering model. In Chat-Captioner, we investigate whether two AI models, unable to individually describe images in detail, can collaborate through an automated, visually guided dialogue to generate a better and more enriched image description than a single AI model. We conduct human-subject evaluations on common image caption datasets such as COCO, Conceptual Caption, and WikiArt, and compare ChatCaptioner with BLIP-2 as well as ground truth. Our results demonstrate that ChatCaptioner's captions are significantly more informative, receiving three times as many votes from human evaluators as BLIP-2 alone for providing the most image information. Besides, ChatCaptioner identifies 53% more objects within the image than BLIP-2 alone measured by WordNet synset matching. Code is available at `https://github.com/Vision-CAIR/ChatCaptioner`.

## 1 Introduction

Asking good questions is not only an essential component of effectively acquiring knowledge, but also plays a pivotal role in enhancing our intelligence and expanding our understanding of the world. Taking medical diagnoses as an example, doctors must ask patients targeted questions about their symptoms to gather relevant information and make accurate diagnoses. Likewise, in scientific research, asking insightful questions is paramount to advancing knowledge and discovering new findings that may have far-reaching implications.

However, the primary focus in recent AI research has been on developing models that can better answer questions, like InstructGPT (Ouyang et al., 2022) in Open-Domain Question Answering (Yang et al., 2015; Rajpurkar et al., 2016; Joshi et al., 2017) and BLIP-2 (Li et al., 2023) in Visual Question Answering (Antol et al., 2015; Goyal et al., 2017; Hudson & Manning, 2019). Despite the significant progress in the question-answering models, their effectiveness in providing useful information is heavily reliant on the quality of the questions they receive. In essence, these models depend on humans to ask insightful questions that can direct their generation of informative answers. If we have an automatic questioning machine that keeps asking informative questions, the human questioners can be replaced and the question-answering models can be guided to provide more valuable knowledge automatically.

---

[*]Equal contribution

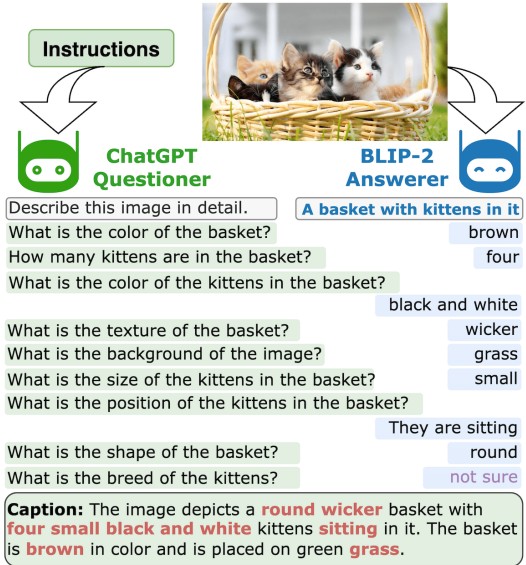

Figure 1: **Example of the dialog between ChatGPT and BLIP-2:** BLIP-2 fails to provide a detailed description in the first message exchange. More details about the image highlighted in red are obtained through multiple conversational interactions between the two models. Eventually, the questioner is able to produce a *more detailed* caption about the image by focusing on multiple aspects of the image.

Recent studies (Wei et al., 2022a; Ouyang et al., 2022; Wei et al., 2020; Kojima et al.) have highlighted the impressive zero-shot learning abilities of Large Language Models (LLMs) that are fine-tuned to follow instructions. These LLMs can perform new tasks in a zero-shot manner when presented with well-crafted instruction prompts. We discover that such LLMs like ChatGPT (OpenAI, 2022) have the ability to keep asking new and contextually relevant questions when properly designed prompts are given. With this capability in place, building an effective automatic questioning machine is now a feasible task.

Based on our findings, we design an automatic questioning system on ChatGPT and integrate it into image captioning, where strong vision-language models like BLIP-2 (Li et al., 2023) are available to answer image-related questions. Our method, named ChatCaptioner, generates more informative and enriched image captions by asking relevant questions to incrementally gain more information. In detail, we design a prompting system that encourages ChatGPT to ask a series of informative questions that maximize its knowledge of the image, building on previous questions and answers. Note that ChatGPT is a pure language model and cannot "see" any visual information. We present the inquired image to BLIP-2 and set it as the question answerer. At the end of the conversation, ChatGPT is prompted to summarize the discussion into a few sentences as the final enriched image description. An example of the conversation between ChatGPT and BLIP-2 and the final caption is shown in Fig.1.

In our experiments, we aim to investigate whether ChatCaptioner, an automated visual dialogue system between a large language model ChatGPT and a visual language model BLIP-2, can acquire more visual information compared to the output of the visual language model alone. We evaluate ChatCaptioner's captions on sampled images from COCO (Lin et al., 2014), WikiArt (Saleh & Elgammal, 2015), and CC (Sharma et al., 2018) datasets based on the human subject evaluation experiment. Compared to BLIP-2's state-of-the-art direct captioning performance, ChatCaptioner receives three times as many votes from human evaluators for providing richer image information. Besides, ChatCaptioner identifies 53% more objects than BLIP-2 alone within the image. Results verify the benefit of good questions to acquire more knowledge from existing AI models and the effectiveness of modern LLMs to serve as zero-shot automatic questioners.

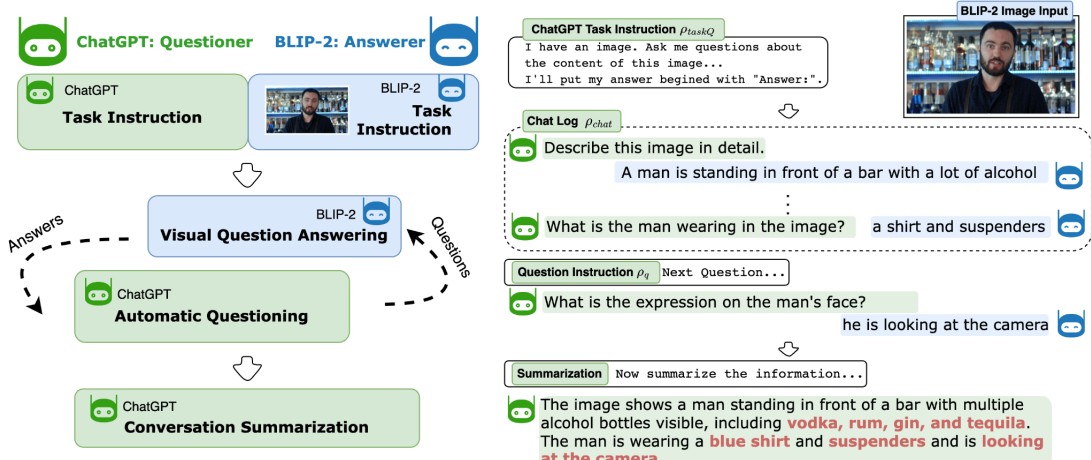

Figure 2: ChatCaptioner contains a questioner ChatGPT and an answerer BLIP-2. We start with a task instruction to set the context and outline the problem for ChatGPT. Once the answer from BLIP-2 is received, we prompt the chat log and question instruction to ChatGPT to continue asking information-acquiring questions. Finally, we provide a summarization instruction to ChatGPT to conclude the conversation as the image captions. Enriched details are highlighted in red.

## 2 Related Works

**Learning to Ask Questions**   There is vast work in NLP community that focuses on question generation. Question generation (Mostow & Chen, 2009; Heilman & Smith, 2010) is the task of generating a question from a given passage and an answer. Sun et al. (2018); Duan et al. (2017); Sun et al. (2023) have concentrated on the generation of question words such as 'when', 'how', and 'why'. Zhao et al. (2018); Tuan et al. (2020) considered leveraging paragraph-level context to enhance question relevance and coherence. Recent methods like Jia et al. (2020); Xiao et al. (2020); Liu et al. (2019); Ghanem et al. (2022); Wang et al. (2022a); Liu et al. (2020); Sultan et al. (2020); Wang et al. (2020); Gou et al. (2023); Nema et al. (2019) have explored different neural network architectures and training strategies for better performance. Tang et al. (2018) and Wang et al. (2022c) have explored question generation within collaborative settings, suggesting a broad spectrum of approaches to refine and diversify question generation methodologies.

However, in cases when we do not have the answer and need to ask questions for the answers, such methods are not applicable. Visual Question Generation (Mostafazadeh et al., 2016; Zhang et al., 2016) is a task aimed at generating natural and engaging questions for a given image. Several works like Patro et al. (2018; 2020); Li et al. (2018); Jain et al. (2017); Vedd et al. (2021); Shen et al. (2019) have been proposed to solve this task. They focus on generating independent questions only and do not have the ability to keep asking new questions based on the previous questions. Our work differs from previous studies significantly. First, we focus on acquiring more knowledge via the generated questions, instead of just generating them. Secondly, our method can keep asking new and relevant questions based on previous questions. Third, our approach leverages modern large language models and requires zero training for questioning.

**Large Language Model and Prompting**   Recent research (Brown et al., 2020; Kojima et al.; Wei et al., 2020; 2022b;a; Chung et al., 2022; Ouyang et al., 2022) has revealed the abilities of Large Language Models (LLMs) like GPT-3 (Brown et al., 2020) or PaLM (Chowdhery et al., 2022) to solve versatile tasks specified by prompting. For example, GPT-3 (Brown et al., 2020) shows the capability to learn new tasks by providing a few task examples provided in the prompt, named in-context learning. Moreover, Chain-of-Thought methods (Kojima et al.; Wei et al., 2020) demonstrate that explicitly asking LLM to solve tasks step-by-step in the prompt improves the performance significantly. Additionally, FLAN (Wei et al., 2022a; Chung et al., 2022) demonstrates that LLMs with instruction tuning can accomplish new tasks in a zero-shot manner. Zhou et al.

(2022) proposes the Automatic Prompt Engineer (APE) by utilizing LLMs to generate instruction-following zero- and few-shot prompts. Further studies, including InstructGPT (Ouyang et al., 2022) and ChatGPT (OpenAI, 2022), show that the performance of LLMs can be enhanced even further by using reinforcement learning from human feedback (Christiano et al., 2017; Stiennon et al., 2020). In our work, we leverage the instruction-following ability of LLMs and design prompts that enable ChatGPT to keep asking new questions about images.

**Image Captioning and Visual Question Answering**  Several works (Lin et al., 2014; Antol et al., 2015; Sharma et al., 2018; Young et al., 2014; Das et al., 2017; de Vries et al., 2017)) have been curated to explore the link between visual and linguistic information, facilitating the investigation of tasks such as image captioning, visual question answering (VQA), and visual dialog. Recent research in vision and language pertaining (Chen et al., 2022; Tsimpoukelli et al., 2021; Alayrac et al., 2022; Wang et al., 2022b; Li et al., 2022; 2023) has advanced the performance for image captioning and visual question answering (VQA) by a large margin. For example, VisualGPT (Chen et al., 2022) shows the benefits of initialization with pretrained language models for more data-efficient training. Frozen (Tsimpoukelli et al., 2021) extend it by finetuning a vision encoder and aligning it with a frozen LLM. BEiT-3 (Wang et al., 2022b) and BLIP (Li et al., 2022) pretrain models using unified transformer architecture. Flamingo (Alayrac et al., 2022) proposes a cross-attention design to align vision and language. BLIP-2 (Li et al., 2023) introduces a lightweight Q-Former that converts visual features into tokens that can be directly understood by a frozen LLM, and achieves impressive results on both image captioning and VQA tasks. In our work, our automatic questioning mechanism leverages the VQA capability of BLIP-2 to extract additional image information and enrich the image captions beyond the original BLIP-2 captions.

## 3   ChatCaptioner

In ChatCaptioner, we design an automatic questioning mechanism based on ChatGPT's zero-shot instruction-following ability to keep asking informative questions about images. BLIP-2, the vision-language model, then provides new image information according to the asked questions. Finally, ChatGPT is prompted to summarize the chat history and generate the final image captions with rich details. An overview of our method is demonstrated in Fig.2.

### 3.1   Automatic Questioning

To activate the questioning ability of ChatGPT, we design a prompting system that enables ChatGPT to generate questions based on previous chat logs. Our prompting system for ChatGPT contains three components: a task instruction for explaining the task $\rho_{taskQ}$, a chat log to store previous questions and answers $\rho_{chat}$, a question instruction for generating high-quality questions $\rho_q$. Each question is generated given the context $\rho_{taskQ}+\rho_{chat}+\rho_q$. In addition, we design a question-trimming mechanism for the automatic post-processing of the generated questions.

**ChatGPT Task Instruction** $\rho_{taskQ}$   The task instruction $\rho_{taskQ}$ sets the context and outlines the task that ChatGPT is required to perform. $\rho_{taskQ}$ directs ChatGPT to generate questions that aim to extract as much information as possible about an image. $\rho_{taskQ}$ is designed as follows:

*I have an image. Ask me questions about the content of this image. Carefully asking me informative questions to maximize your information about this image content. Each time ask one question only without giving an answer. Avoid asking yes/no questions. I'll put my answer beginning with "Answer:".*

The prompt "Each time ask one question only without giving an answer" in $\rho_{taskQ}$ is designed to instruct ChatGPT to generate only one question per round. Without this prompt, ChatGPT may generate a set of questions at once, rather than generating them one by one in response to each answer.

**Chat Log** $\rho_{chat}$   After the task instruction $\rho_{taskQ}$, we include a chat log $\rho_{chat}$ containing all the previous questions and answers. $\rho_{chat}$ follows a template for each Question-Answer round, which is as follows:

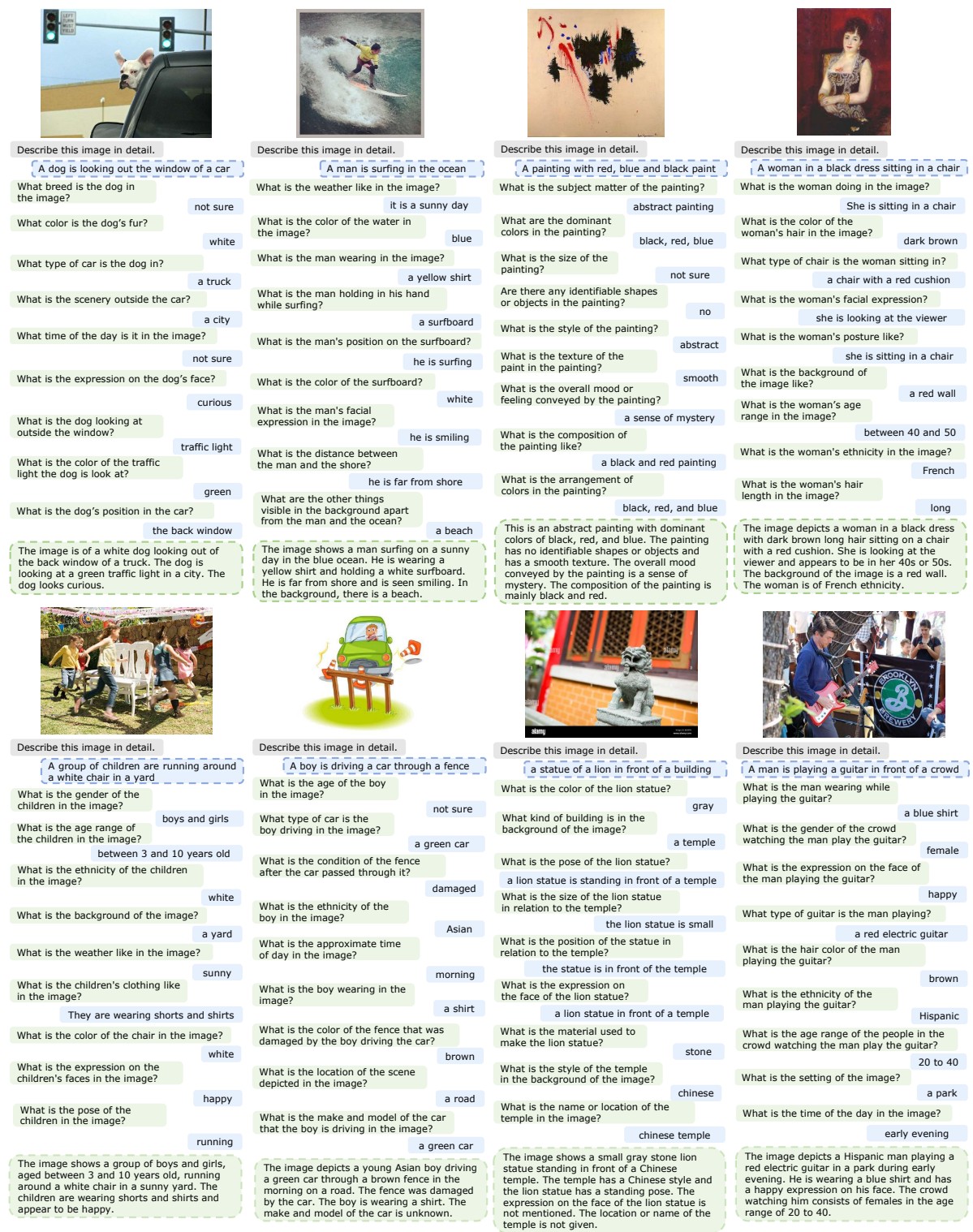

Figure 3: Qualitative examples of the chat log and the captions from ChatCaptioner in various images from COCO (Lin et al., 2014), WikiArt (Saleh & Elgammal, 2015), and CC (Sharma et al., 2018). Compared to the initial description from BLIP-2, questions from ChatGPT extract additional image information and lead to enriched final captions.

*Question: ⟨question⟩ Answer: ⟨answer⟩*

Note that we hard-code the first question as "Describe the image in detail" to start the conversation. Based on the first response of BLIP-2, which provides a brief initial description of the image, ChatGPT is prompted to ask follow-up questions to extract more information about the image.

**Question Instruction $\rho_q$**   To guide ChatGPT in generating new questions, we provide a question instruction $\rho_q$ before each question generation. $\rho_q$, located after the chat log, cues ChatGPT to generate a new question and aims to ensure that the questions are of high quality. It's designed as follows:

*Next Question. Avoid asking yes/no questions. Question:*

The prompt "Next Question" in $\rho_q$ is critical to ensure that ChatGPT continues to ask questions. Without it, ChatGPT may produce undesired sentences after a few Question-Answer rounds. Additionally, we notice that ChatGPT prefers asking yes/no questions which are usually not as informative as other questions. We therefore add the prompt "Avoid asking yes/no questions" to reduce the generation of yes/no questions.

**Question Trimming**   Despite our explicit instruction to not answer the question itself, we observe that sometimes ChatGPT fabricates an answer after asking the question. Fortunately, we find that these fabricated answers always begin with the text "Answer:", following the template specified in the prompt. Therefore, we automatically remove these fabricated answers by discarding the generated text starting from "Answer:".

## 3.2   Question Answering

Similar to ChatGPT, our BLIP-2 prompting mechanism consists of three components: a task instruction $\rho_{taskA}$, the chat log $\rho_{chat}$ same as the ChatGPT one, and an answer instruction $\rho_a$. Each answer generation is prompted by $\rho_{taskA} + \rho_{chat} + \rho_a$. Also, we have an answer-trimming mechanism for post-processing.

**BLIP-2 Task Instruction $\rho_{taskA}$**   We design the BLIP-2 task instruction $\rho_{taskA}$ to alleviate the issue of hallucinating non-existent information in the image. $\rho_{taskA}$ includes an uncertainty prompt "If you are not sure about the answer, say you don't know honestly" that encourages BLIP-2's honest admission of lack of knowledge. The instruction is as follows:

*Answer given questions. If you are not sure about the answer, say you don't know honestly. Don't imagine any contents that are not in the image.*

**Answer Instruction $\rho_a$**   After the chat log $\rho_{chat}$, we provide a straightforward answer instruction to guide BLIP-2's answering process. The instruction is structured as follows:

*Question: ⟨question⟩ Answer:*

**Answer Trimming**   Similar to ChatGPT, BLIP-2 occasionally generates a question after providing an answer. As the LLM backend of BLIP-2, the FLAN-T5 model (Chung et al., 2022), has a much weaker questioning ability than ChatGPT shown later in the question analysis in the experiment section, we automatically filter out these questions by discarding any texts starting with "Question:".

## 3.3   Context Summarizing

To obtain a concise summary of the conversation between ChatGPT and BLIP-2 as the final image caption, we use a summarization instruction after the conversation. This instruction, located after the chat log, prompts ChatGPT to generate a summary using the following structure:

*Now summarize the information you get in a few sentences. Ignore the questions with answers no or not sure. Don't add information. Don't miss information. Summary:*

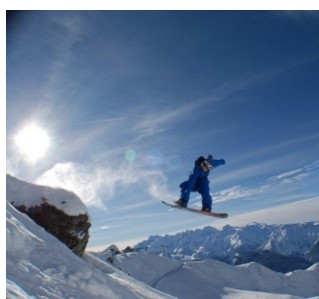

**BLIP-2**:
A person is snowboarding down a mountain
**ChatCaptioner**:
The image depicts a male snowboarder **wearing a blue jacket and pants**, snowboarding down a **snowy** mountain under **sunny weather**. The snowboard and the mountain are both blue and the person is **in the air holding the snowboard**.

| | CIDEr | ROUGE | METEOR |
|---|---|---|---|
| **BLIP-2** | 83.7 | 61.1 | 24.5 |
| **ChatCaptioner** | 0.0 | 23.9 | 18.7 |

Figure 4: An example of the limitation of traditional image caption metrics. Although ChatCaptioner extracts more image details highlighted in red compared to BLIP-2, it receives much lower scores under traditional metrics.

## 4 Experiments

We explore ChatCaptioner from various perspectives through a series of experiments, including an assessment of the informativeness and accuracy of its captions, as well as an analysis of the quality of its generated questions. Qualitative results including chat logs and final captions on various images from different datasets are shown in Fig.3.

**Details of Model Deployment.** For our experiments, we use the ChatGPT model *"gpt-3.5-turbo"* available on the OpenAI API (OpenAI, 2023a). This model is the most powerful GPT-3.5 model accessible through the API during our project. For BLIP-2, we use the biggest version containing a FLAN-T5 (Chung et al., 2022) language model with 11 billion parameters and a ViT-G/14 model from EVA-CLIP (Fang et al., 2022). In all experiments, BLIP-2 answers 10 questions per image, with the first question being hard-coded as *"Describe the image in detail."*. The remaining 9 questions are from ChatGPT, unless otherwise specified.

**Limitation of Traditional Metrics.** The conventional image captioning metrics such as(Vedantam et al., 2015), ROUGE (Lin & Hovy, 2002), and METEOR (Banerjee & Lavie, 2005) are frequently employed to measure the quality of captions. However, the usage of these metrics to evaluate ChatCaptioner can be limiting, because these metrics are designed to measure the similarity between the tested caption and reference captions, assuming that the reference captions are perfect image descriptions. Since ChatCaptioner generates captions that contain a greater level of detail than the reference captions, the metrics may yield low similarity scores, leading to inferior overall performance scores compared to other image captioning approaches like BLIP-2. This issue is depicted in Fig.4. Thus, in our experiments, we primarily rely on human assessments to comprehensively analyze the performance of ChatCaptioner from various perspectives. The design of all the human evaluation interfaces is presented in the supplementary.

### 4.1 Information Analysis

**Does ChatCaptioner extract more information from the image?** We design an experiment to evaluate whether ChatCaptioner is able to generate captions with more information about the image than BLIP-2 alone and ground truth captions. We randomly selected 100 photos from the COCO (Lin et al., 2014) validation set, 100 artworks from WikiArt (Saleh & Elgammal, 2015) dataset with ground truth captions from ArtEmis (Achlioptas et al., 2021), 100 internet images from the Conceptual Captions (CC) (Sharma et al., 2018) validation dataset, and 100 images with detailed and long ground truth captions from the Open Image

Table 1: Human votes on the captions containing the most image information.

| Methods | COCO | WikiArt | CC | OI-LN | Avg. |
|---|---|---|---|---|---|
| GT | 26% | 14% | 8.5% | 33.5% | 20.5% |
| BLIP-2 | 21% | 12.5% | 23% | 6.5% | 15.8% |
| **Ours** | **53%** | **73.5%** | **68.5%** | **63.8%** | **65%** |

Table 2: Numbers of objects discovered by captions.

| Methods | Covered/All | Ratio | Improved |
|---------|-------------|-------|----------|
| BLIP-2 | 383/1154 | 33.2% | - |
| **Ours** | **586**/1154 | **50.8%** | **53.0%** |

Table 3: Correctness Analysis. BLIP-2 correctly answers 66.7% of ChatGPT's questions. 81% of the final captions are deemed correct by humans. Besides, 94% of the wrong captions are caused by BLIP-2's wrong answers.

| | COCO | WikiArt | CC | Avg. |
|---|------|---------|-----|------|
| Answer Correct Rate | 64% | 73% | 63% | 66.7% |
| Caption Correct Rate | 77% | 78% | 88% | 81% |
| Issues From BLIP-2 | 100% | 82% | 100% | 94% |

Localized Narratives (OI-LN) (Pont-Tuset et al., 2020) dataset. Human evaluators on Amazon Mechanical Turk are presented with an image and four captions - one from our method, one from BLIP-2, one ground truth caption, and one fake caption for quality control. Evaluators are asked to pick the caption that offers the richest information about the image. Results are demonstrated in Tab.1. On average, ChatCaptioner receives three to four times as many votes as pure BLIP-2's captions and ground truth captions, showing that by combining two AI models via questioning and asking, the system can provide more detailed image description than a single AI model alone.

**How many objects in images can ChatCaptioner discover?** We randomly sampled 200 images from Pascal VOC (Everingham et al., 2010) and considered all class labels in the segmentation masks as the ground truth objects. We then assessed how many of these objects are included in the captions. We utilize WordNet from NLTK (Bird et al., 2009) to find words with similar semantic meanings based on the Wu-Palmer Similarity of their synsets. Tab.2 presents the experimental results, where 1154 objects are identified in the 200 sampled images. BLIP-2 covers only 383 of them, while ChatCaptioner increases the coverage by 53% to 586, suggesting that the automatic questioning helps BLIP-2 find more objects in the images.

## 4.2 Correctness Analysis

**How accurate are the captions from ChatCaptioner?** We conducted a human evaluation where evaluators were presented with an image and a generated caption, as well as all Q&A between ChatGPT and BLIP-2. The evaluators need to verify the correctness of the caption, select incorrect answers from BLIP-2, and judge whether the incorrectness can be attributed to the wrong answers. The experiments were performed on samples from COCO (Lin et al., 2014), WikiArt (Saleh & Elgammal, 2015), and CC (Sharma et al., 2018) datasets. Each image was evaluated by 4 different evaluators. Results presented in Tab.3 reveal that about 80% of the captions are deemed correct. Moreover, BLIP-2 answers around 67% of the questions correctly. Among the incorrect captions, 94% are caused by BLIP-2's wrong answers, suggesting that BLIP-2 is the primary source of incorrectness. This implies that using a more powerful VQA model may help to enhance the overall performance of the system in the future.

**Does BLIP-2 know it doesn't know?** BLIP-2 usually makes up answers if the question cannot be answered based on the image. In other words, BLIP-2 doesn't know that it doesn't know this information. To mitigate this issue, we incorporate an uncertainty prompt *"If you are not sure about the answer, say you don't know honestly."* in our BLIP-2 task instruction $\rho_{taskA}$. Two examples showing the effectiveness of the uncertainty prompt are demonstrated in Fig.5 and more examples are in Fig.3 and the supplementary.

**How effective is the uncertainty prompt?** We randomly selected 200 images from the CC (Sharma et al., 2018) and collected 1,800 questions based on these images. We then identify 147 questions that BLIP-2 is uncertain about, present these questions to human evaluators, and ask them to answer based on the image content. Results in Tab.4 demonstrate that approximately 60% of these questions are deemed unanswerable based on the image. For the remaining answerable questions, BLIP-2 cannot correctly answer 30 of them.

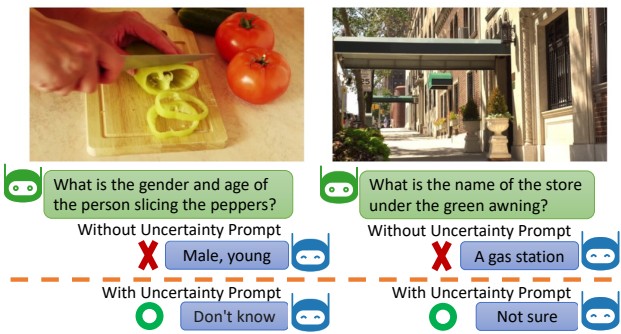

Figure 5: Examples of BLIP-2's answers with and without the uncertainty prompt. The uncertainty prompt helps BLIP-2 avoid making up an answer when it encounters questions that cannot be answered based on the image.

Table 4: Analysis on questions that BLIP-2 is unsure about. 60% deemed unanswerable by humans. 20% cannot be correctly answered by BLIP-2. Overall, BLIP-2 makes mistakes on 80% of these questions.

|  | Total Uncertain Questions | Unanswerable Questions | Answerable But Wrong | Avoided Bad Answers |
|---|---|---|---|---|
| Num. | 147 | 88 | 30 | 118 |
| Ratio | - | 60% | 20% | 80% |

In total, without the uncertainty prompt, BLIP-2 will generate 118 incorrect answers out of 147 uncertain questions, resulting in an error rate of approximately 80%. In addition, out of the original 1800 questions, BLIP-2 has 674 wrong answers. Taking the 147 potential wrong answers avoided by the uncertainty prompt into account, the uncertainty prompt reduces about 15% of the wrong answers.

## 4.3 Question Analysis

**How diverse are the automatically generated questions?** We analyze 1419 unique questions filtered from 1800 generated questions on 200 random CC (Sharma et al., 2018) samples. To visualize the diversity

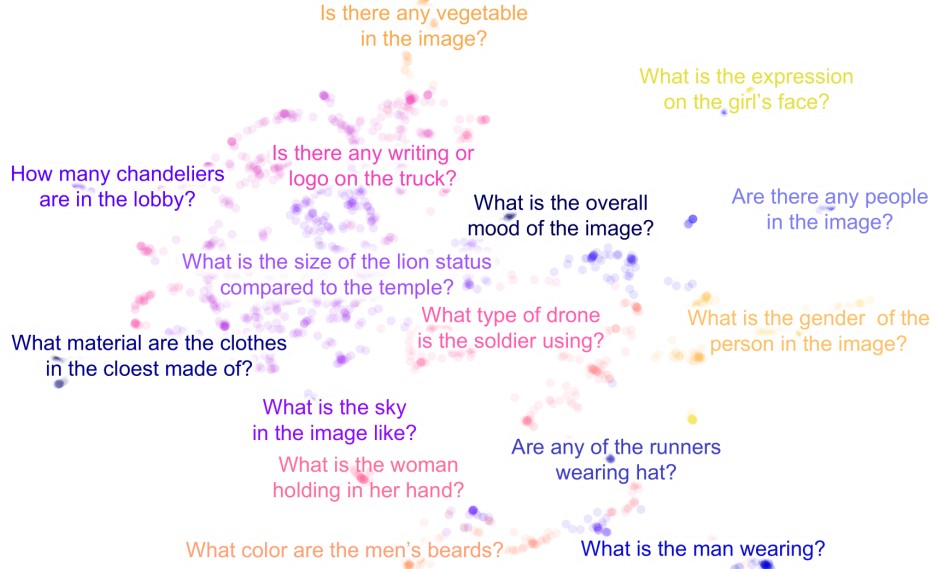

Figure 6: Visualization of question embeddings and question examples printed at the corresponding positions. Our method can ask diverse questions focusing on various perspectives of the image.

Table 5: Number of unique questions. InstructGPT and ChatGPT excel at generating diverse questions and rarely repeating questions within a dialogue.

| Unique Q/Total Q | OPT 6.7B | FLAN-T5 | InstructGPT | ChatGPT |
|---|---|---|---|---|
| Per Dialogue | 1.75/9 | 2.03/9 | 9/9 | 8.98/9 |
| All Questions | 166/1800 | 169/1800 | 1400/1800 | 1419/1800 |

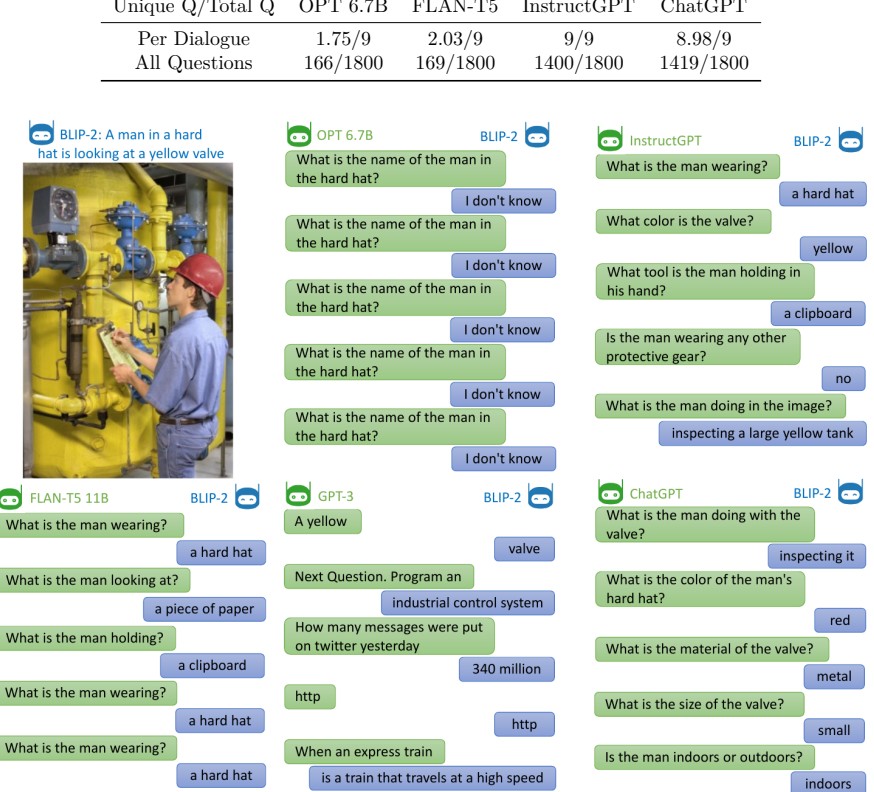

Figure 7: Question examples from various LLMs. ChatGPT and InstructGPT are able to ask new and informative questions. FLAN-T5 11B and OPT 6.7B tend to repeat old questions. GPT-3 fails to ask any related questions.

of the questions, we compute the question embedding based on Sentence-BERT (Reimers & Gurevych, 2019) and reduce the dimension to two for visualization via UMAP (McInnes et al., 2018). The resulting visualization is presented in Fig.6.

Our analysis shows that the questions from ChatCaptioner are diverse and cover various perspectives, including high-level information like *"What is the overall mood of this image?"*, object properties like *"What material are the clothes in the closet made of?"*, environmental information like *"What is the sky in the image like?"*, and so on. This suggests that our automatic questioning mechanism is able to generate diverse questions, explore various aspects of the image, and acquire more informative responses.

**Can we use other LLMs as questioners?** In addition to ChatGPT, we evaluate the automatic questioning capabilities of several other large language models (LLMs), including InstructGPT (Ouyang et al., 2022), GPT-3 (Brown et al., 2020) without instruction finetuning, FLAN-T5 11B (Chung et al., 2022) (11B parameters), and OPT 6.7B (Zhang et al., 2022) (6.7B parameters). Fig.7 showcases the different question-asking abilities given the initial description from BLIP-2. We observe that InstructGPT shows a similar question-asking ability as ChatGPT and is able to generate meaningful and diverse questions. While FLAN-T5 11B and OPT 6.7B also exhibits the ability to ask questions, it repeats a single question several times. Interestingly, GPT-3 fails to generate any meaningful questions related to the image.

We further analyze their 1800 generated questions on 200 CC (Sharma et al., 2018) samples. We skip the poorly performed GPT-3 here. LLMs' questioning abilities are evaluated in two ways. Firstly, we measure the average number of unique questions per image, which indicates whether the model can keep asking new

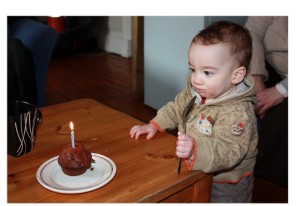

**ChatCaptioner (ChatGPT+BLIP-2):**
In the image, a baby is standing in front of a wooden table. The baby is wearing a brown hoodie and appears to be curious. On the table, there is a small brown cupcake with a white candle. The baby is holding a fork and the cupcake, but is not looking directly at the camera.
**GPT-4V:**
The image shows a young child staring at a small lit birthday candle on top of what appears to be a muffin or a small cake. The child is holding a fork and is standing close to a wooden table with the cake placed on a white plate. There's a mug to the left side of the plate, and you can see part of another person to the right side of the photo, though they are mostly out of frame. The scene suggests a birthday celebration, and the child seems curious or thoughtful about the lit candle.

Figure 8: An qualitative comparision between GPT-4V and ChatCaptioner (ChatGPT + BLIP-2).

Table 6: Ablation study with the GPT-4V answerer.

| Questioner | Answerer | Human Votes |
|---|---|---|
| ChatGPT | GPT-4V | 92% |
| ChatGPT | BLIP-2 | 8% |

questions in a single dialogue. Secondly, we count the total unique questions to see if the questioner could customize the questions according to the given contexts or just ask fixed predefined questions.

Results in Tab.5 reveal that ChatGPT and InstructGPT almost never repeat their question in a single dialogue and generate around 1400 unique questions out of 1800, suggesting that they are able to ask new questions according to the contexts. In contrast, FLAN-T5 11B and OPT 6.7B tend to repeat old questions, averaging about only 2 unique questions per image and generating less than 170 unique questions in total. Our study suggests that to develop a questioning machine that can generate novel and customized questions, it may be necessary to utilize instruction-finetuned LLMs with dozens of billions of parameters.

## 4.4 Exploration with GPT-4V

**How are the results when combined with GPT-4V?** With the rapid advancement in this field, advanced vision-language models with robust capabilities for detailed image captioning, such as GPT-4VOpenAI (2023b), are now accessible. To evaluate the caption quality between ChatCaptioner (ChatGPT+BLIP-2) and GPT-4V, we randomly selected 20 COCO images and instructed human annotators to choose their preferred captions. The results indicate a unanimous preference for captions generated by GPT-4V in all 20 instances. A qualitative example is provided in Figure 8. Generally, captions from GPT-4V encompass a greater number of objects and location details, illustrating features such as "There's a mug to the left side of the plate", in comparison to those produced by ChatCaptioner (ChatGPT+BLIP-2).

**Can a better vision answerer improve ChatCaptioner?** In this study, we substituted GPT-4V for BLIP-2 as the answerer, while retaining the original questioner. The experimental results, detailed in Table 6, reveal that human annotators prefer the pairing of the ChatGPT questioner with the GPT-4V answerer over the original ChatGPT questioner and BLIP-2 answerer in 92% of instances. This preference highlights the significance of a proficient answerer within our system. The suboptimal performance of the ChatCaptioner system when paired with the BLIP-2 answerer may be attributable to the incorrect responses provided by BLIP-2, as discussed in Section 4.2.

**Can ChatCaptioner further improve GPT-4V?** ChatCaptioner is an automated question-answering system designed to extract more information from images than a single model alone. The design of Chat-Captioner is orthogonal to the choice of underlying models. In our experiment, we integrated ChatCaptioner

Table 7: Human votes on the captions containing the most image information.

| Methods | GPT-4V alone | GPT-4V + ChatCaptioner |
|---|---|---|
| Votes | 3% | 97% |

with GPT-4V, where GPT-4V serves as both the questioner and answerer within the ChatCaptioner framework, with all other settings remaining constant. Human evaluators were tasked with selecting captions that most accurately capture the details of an image, comparing the standalone output of GPT-4V against its performance within the ChatCaptioner framework. The results, presented in Table 7, indicate that in 97% of cases, captions produced by GPT-4V combined with ChatCaptioner contain more detailed image information, thus confirming the effectiveness of our proposed framework in eliciting additional details through its question-and-answer mechanism.

## 5  Conclusion

In this work, we discover that advanced large language models possess the ability to pose insightful and diverse questions when provided with well-crafted prompts. Based on our findings, we develop an automatic questioning system named ChatCaptioner for the task of image captioning. By prompting ChatGPT to keep asking questions that expand its understanding of an image, ChatCaptioner guides BLIP-2 to provide comprehensive image information, resulting in image captions that are significantly more detailed and enriched than those from BLIP-2 alone. ChatCaptioner demonstrates the power of automatic questioning systems to effectively extract desired information. Through our work, we aim to draw attention to the potential of automatic questioning systems in AI and inspire further research in various domains.

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

# A  Appendix

## A.1  Cost

Our method is based on the ChatGPT model, specifically the *gpt-3.5-turbo* version which we access through OpenAI's API. At the time of our project, the cost for using 1000 tokens in *gpt-3.5-turbo* was 0.002 US Dollars. On average, we spent approximately 2500 tokens for each image for ten Question-Answer rounds, which translates to a cost of approximately 0.005 US Dollars per image.

## A.2  Yes/No Question Ablation

Usually, yes/no questions contain relatively less information. To reduce the generation of yes/no questions from ChatGPT, we explicitly add a prompt *"Avoid asking yes/no questions"* in the task instruction $\rho_{taskQ}$ and the question instruction $\rho_q$. Our ablation study in Tab.8 shows that this prompt reduces the generation of yes/no questions from 33% of the cases to 2% in 1800 questions on 200 random CC Sharma et al. (2018) samples, verifying its effectiveness.

Table 8: Effectiveness of the yes/no prompt.

|       | Total Question | Yes/No Question w/o Prompt | Yes/No Question with Prompt |
|-------|----------------|----------------------------|-----------------------------|
| Num.  | 1800           | 595                        | 38                          |
| Ratio | -              | 33%                        | 2%                          |

## A.3  Using Vicuna as Questioners

In addition to the open-sourced LLMs Flan-T5 11B and OPT 6.7B in the main paper, here we conduct experiments with one of the latest SOTA open-sourced LLMs, Vicuna 13B Chiang et al. (2023), as the questioner and evaluate the generated question quality as shown in Tab.9. Experimental results suggest that Vicuna is able to generate diverse questions like ChatGPT and rarely repeats the questions in a single 9-question dialogue.

Table 9: Number of unique questions.

| Unique Q/Total Q | OPT 6.7B  | FLAN-T5   | Vicuna     | ChatGPT    |
|------------------|-----------|-----------|------------|------------|
| Per Dialogue     | 1.75/9    | 2.03/9    | 8.88/9     | 8.98/9     |
| All Questions    | 166/1800  | 169/1800  | 1710/1800  | 1419/1800  |

## A.4   Human Evaluation Interface

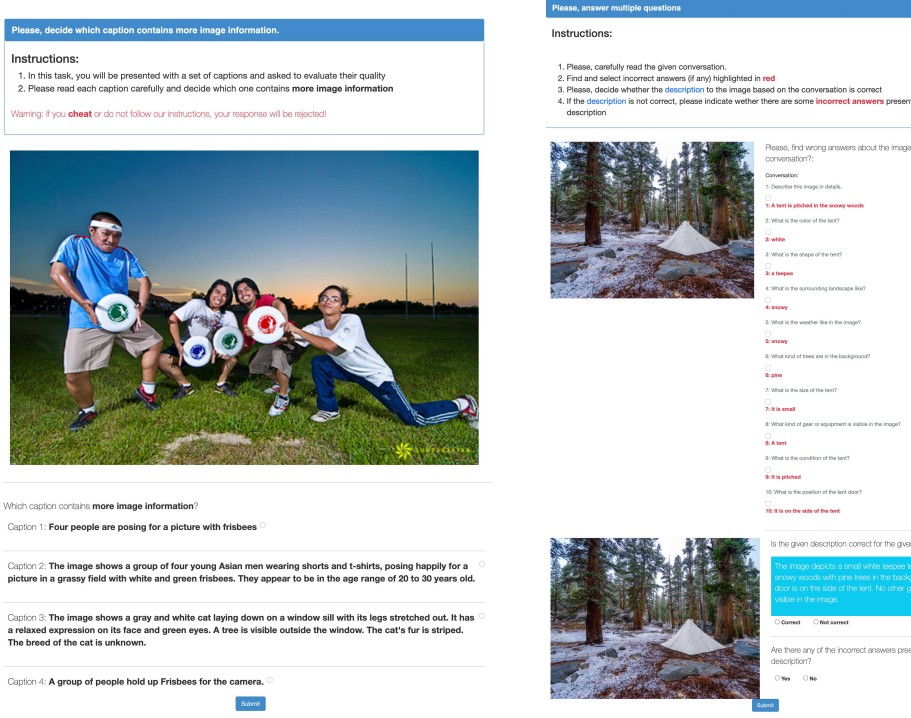

(a) Human evaluation interface of the information experiments.

(b) Human evaluation interface of the correctness experiments.

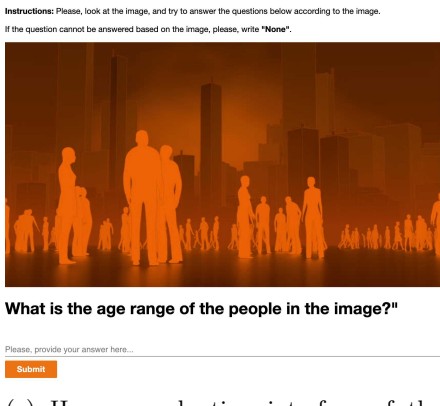

(c) Human evaluation interface of the question-answerable experiments.

Figure 9: Human evaluation interfaces

## A.5 Random Examples of Generated Questions in Conceptual Captioning Dataset

What is the material of the pier in the image?
What is the position of the sign that says "No Swimming Allowed" on the dock?
What is the material of the valve?
What is the color of the plate on which the cake is placed?
What is the expression on the man's face?
What is the boy's expression while playing with the sailboat?
What is the angle of the camera in the image?
What type of flower is it?
What is the color of the woman's glasses?
What is written on the quote on the mountain?
What is the design on the cake?
What is the woman's hair color in the image?
Are the man and woman standing or sitting in the image?
What is the location of the scene depicted in the image?
What is the boy's expression?
What is the material of the pink running shoes?
What is the expression on the man's face?
What type of vegetation surrounds the pond in the image?
What is the size of the fountain in the image?
What is the name of the mountain range seen in the background of the image?
What is the name of the park?
What is the design of the woman's dress?
What is the color of the chainsaw?
What is the ethnicity of the two men in the image?
What is the woman's pose in the photo?
What modifications, if any, have been made to the car in the image?
What kind of donuts are in the box?
What is the woman's age range in the image?
What is the weather like in the image?
What is the man's posture like in the image?
What kind of lighting is in the room?
What is the woman's hair color in the image?
What is the woman wearing in the image?
What is the woman's pose in the image?
What is the type of the lightning bolt? (e.g. cloud-to-ground, cloud-to-cloud)
What is the context or setting of the image?
What type of event is taking place where the man is performing?
What is the shape of the base or stand of the ice sculpture?
What is the size of the tent?
What is the boy wearing?
What is the man doing?
What is the material of the sink in the image?
What is the age range of the people in the image?
What is the style of the hat?
What is the background of the image?
What is the surrounding environment like in the image?
What is the woman wearing?
What is the shape of the backyard?
What is the material of the backpack the man is carrying?
What is the texture of the granny square?
What is the style of the bridge?
What is the size of the sun in the image?

## A.6 More Qualitative Results

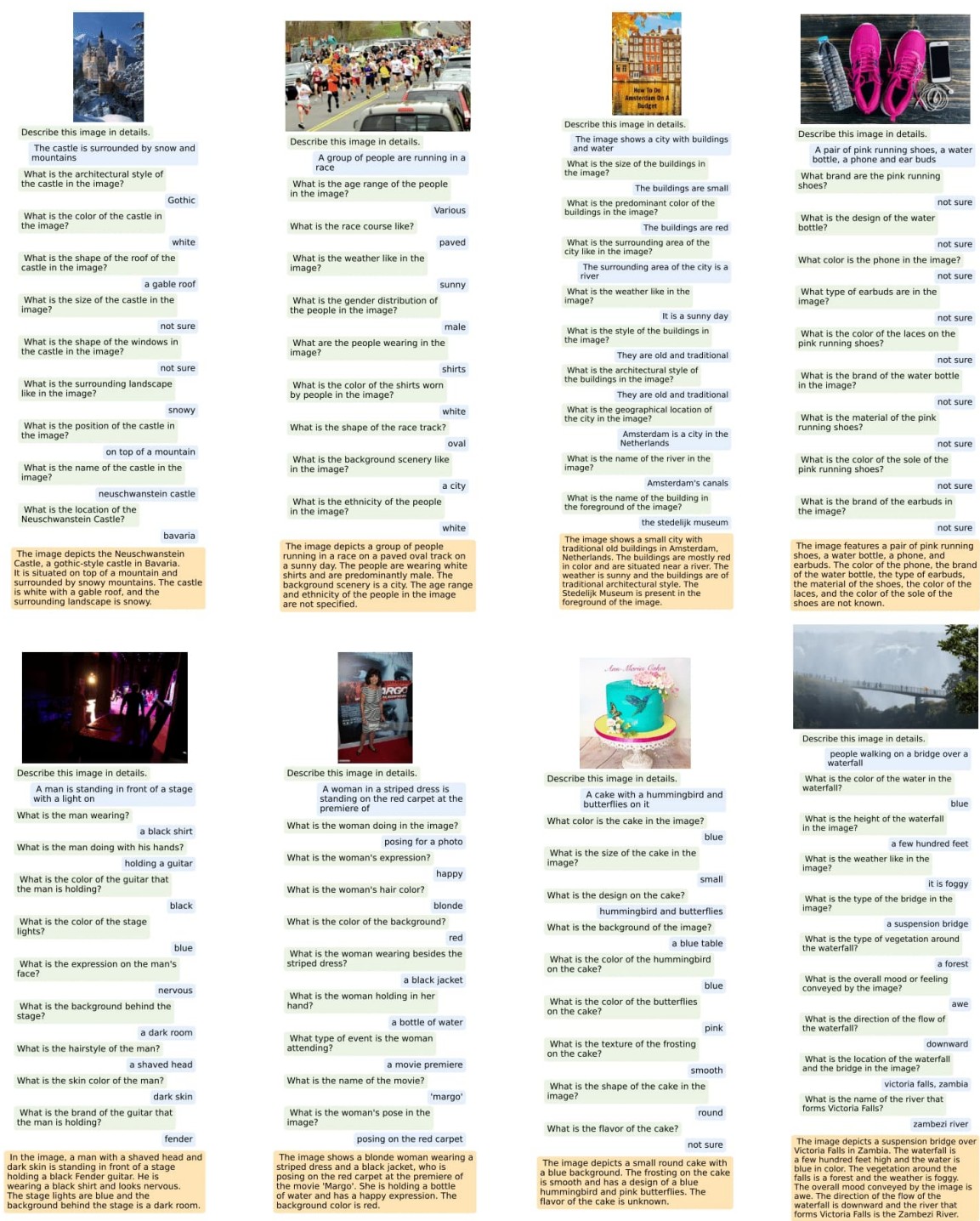

Figure 10: Random samples from CC Sharma et al. (2018).

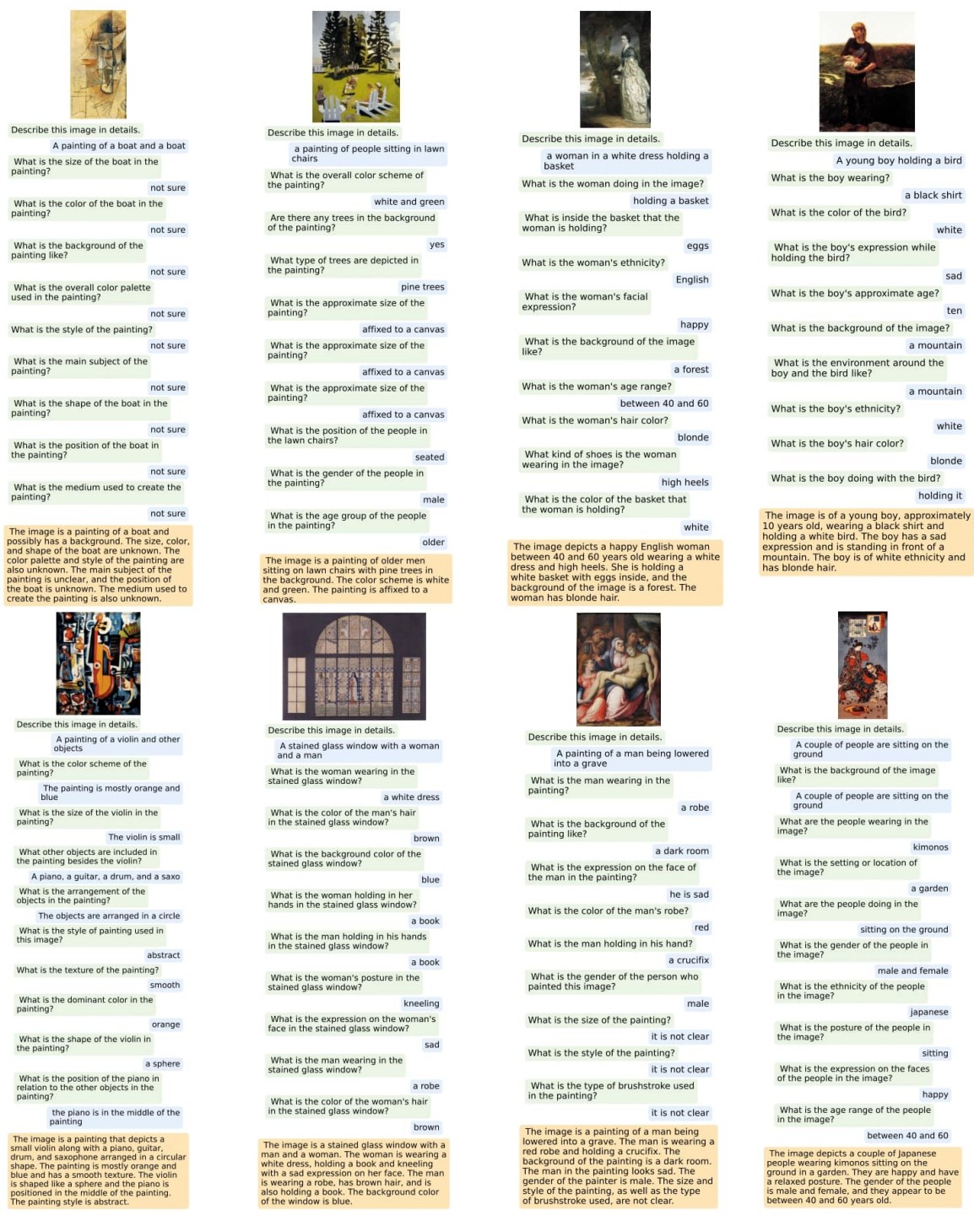

Figure 11: Random samples from WikiArt Saleh & Elgammal (2015).

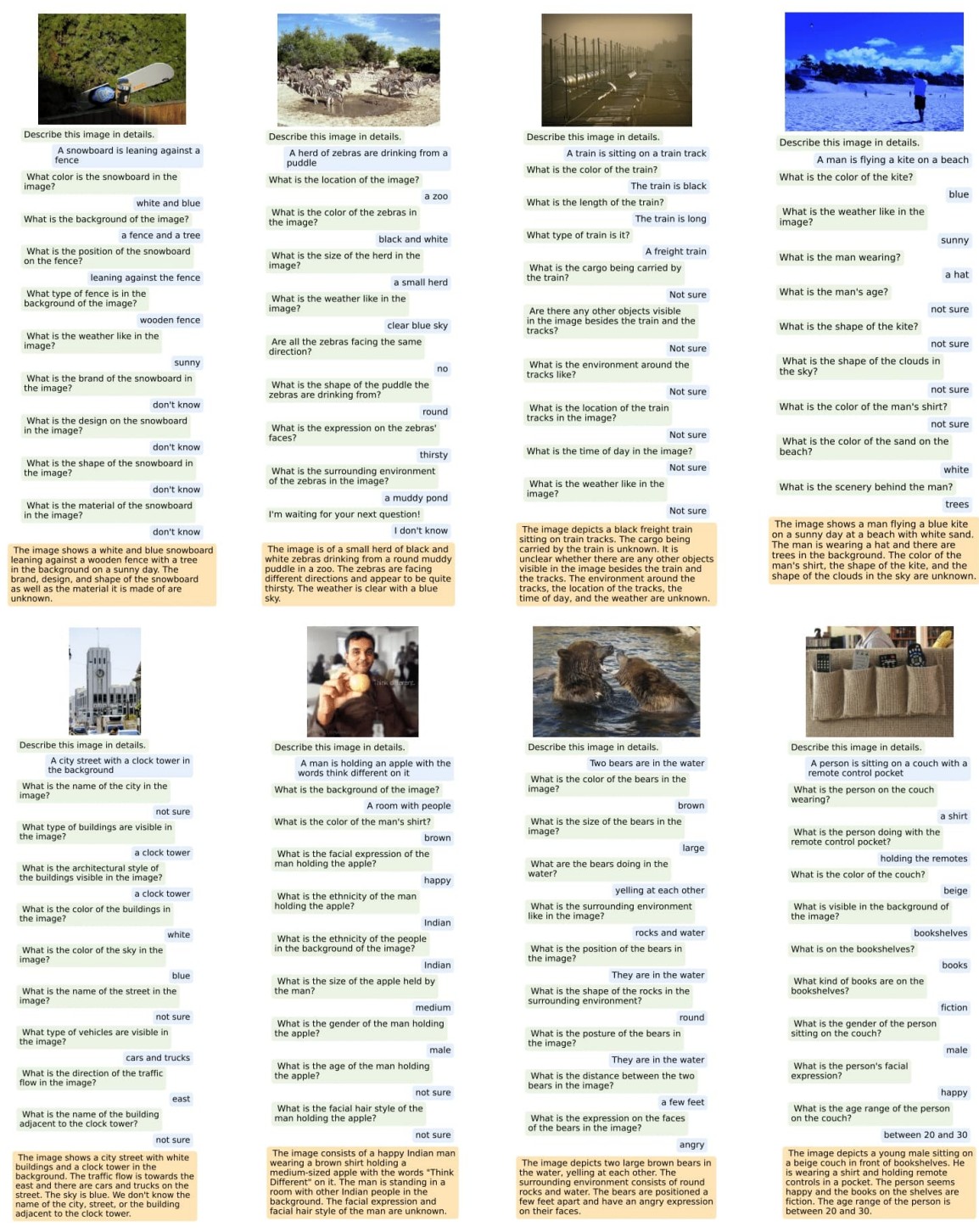

Figure 12: Random samples from COCO Lin et al. (2014).

