# OpenReview forum: "ChatGPT Asks, BLIP-2 Answers: Automatic Questioning Towards Enriched Visual Descriptions"
_TMLR — Accepted by TMLR_

### Review · Reviewer_ekHZ · 2023-12-02

**Summary Of Contributions:**

This paper proposes a new image-captioning method where ChatGPT is used to generate questions that a VL model answers. This dialog is then summarized to form a caption.

**Audience:**

Yes

**Claims And Evidence:**

Yes

**Requested Changes:**

Questions (Minor)
- Given how much longer it appears the ChatCaptioner response is than the BLIP-2 response (5 lines compared to 1 line in Figure 4), how can the BLIP-2 response be voted to contain more information than the ChatCaptioner response?
- In the Correctness Analysis, if the incorrectness was not attributed to the wrong answers, then does it have to be due to the summarization from ChatGPT?

**Strengths And Weaknesses:**

Strengths:
Overall, I liked the paper. The idea is simple, yet interesting and works well. I thought the experiments/analysis was pretty complete (especially given the difficulty of evaluating text generations in general). There were several times I was writing out comments for analysis I thought were missing only to find it later in the paper - how accurate are the captions, effectiveness of the uncertainty prompts. And these analyses all had quantitative results. The paper was well written and easy to follow and the diagrams were easy to follow.

Weakness:
I don’t think there were many weaknesses. One question I had was how do you think this method is situated in light of the release of GPT-4, which has a vision component?

---

> ### Author Response · Authors · 2024-02-05
>
> Thank you for your feedback! Here we answer your questions below
>
> **Q1: how is this method situated in light of the release of GPT-4V?**
>
>
> This paper's primary contribution is an automated questioning-answering conversation system that extracts more image information than a single model alone. Our contribution is perpendicular to the model designs. With more powerful MM-LLMs than BLIP-2 currently available, we conducted a new experiment utilizing GPT-4V, the strongest MM-LLM. In this experiment, GPT-4V functions as both questioner and answerer within the ChatCaptioner framework, while maintaining all other settings. Human evaluators were asked to choose the caption that accurately encompasses more image details, comparing GPT-4V's standalone generation to its application within the ChatCaptioner framework. Results demonstrate that in 97% of cases, captions from GPT-4V+ChatCaptioner provide more image information, validating the efficacy of our proposed framework in extracting further details through questioning and answering.
>
>
>
> | Method         | GPT-4V  | GPT-4V+ChatCaptioner |
> |----------------|---------|----------------------|
> | Human Votes    | 3/100   | 97/100                |
>
>
>
>
> **Q2: In the Correctness Analysis, if the incorrectness was not attributed to the wrong answers, then does it have to be due to the summarization from ChatGPT?**
>
>
> Yes, ChatGPT is not flawless and might produce imperfect summaries. As shown in Table 3 of the main paper, the model performs well in summarizing common photos (from COCO) and internet images (from CC), with all errors in the tested images attributed to BLIP-2. However, ChatGPT's performance declines when dealing with paintings (from WikiArt), resulting in 18% of the errors.
>
>
>
> **Q3: how can the BLIP-2 response be voted to contain more information than the ChatCaptioner response?**
>
>
> The quality of ChatCaptioner's responses relies on the accuracy of BLIP-2's answers. If BLIP-2 delivers incorrect or irrelevant replies to ChatGPT's inquiries, it negatively impacts ChatCaptioner's final response, causing human annotators to favor pure BLIP-2's response or the ground truth instead.

---

> > ### Comment · Reviewer_ekHZ · 2024-02-05
> > **Response to authors**
> >
> > Thank you for answering my questions and running and adding the new GPT-4V results! Given how all the other reviewers mentioned a similar point, I think a discussion around this with the results should be added to the paper since it seems this is a question readers will ask.

---

> > > ### Author Response · Authors · 2024-02-09
> > >
> > > Thank you for the suggestion. We have revised the paper to incorporate the GPT-4V-related experiments highlighted in blue in Section 4.4.

---

> > > > ### Comment · Reviewer_ekHZ · 2024-02-09
> > > > **Response to authors**
> > > >
> > > > Thank you for revising the paper with the new experiments. All my concerns are addressed.

---

### Review · Reviewer_ch9K · 2023-12-10

**Summary Of Contributions:**

The paper presents a system (code is being made available) that will use ChatGPT (or InstructGPT and other related models) to repeatedly ask questions about an image, which BLIP-2 will then "answer" in a dialog. At the end, ChatGPT summarizes the acquired information. Using rule-based prompting, ChatGPT will thus extract more information from an image than a single invocation of BLIP-2 would yield. ChatGPT is using BLIP-2 like a "tool" to extract information, which it then summarizes.
The paper evaluates this idea on a number of datasets (e.g. COCO). Since the output of ChatGPT does not correspond to typical captioning patters (so CIDEr, ROUGE, etc cannot easily be applied), Human evaluation is used, which shows that this process yields more information (e.g. more objects or more properties) with a reasonably high level of "correctness". An "uncertainty" prompt is introduced to improve results and authors present some plots that show the diversity of the generated questions.

**Audience:**

Yes

**Claims And Evidence:**

No

**Requested Changes:**

- There is a body of work in question generation for NLP (diversity, quality, etc), which the paper does not cite - it would be helpful to put this work in context
- Please discuss the impact of recent releases (esp MM-LLMs) on this work -- is this still relevant? Which parts of the results will carry over to other models/ tasks?
- How were the prompts designed? Was there any attempt to optimize the questioning strategy with ChatGPT, or when porting to other LLMs?
As-is, the paper reads a bit like a snap-shot of an interesting & relevant idea, but given the speed with which the field moves forward, and the emergency of multi-modal LLMs, I am wondering if that paper's

**Strengths And Weaknesses:**

Strengths
- The paper makes a good point in arguing that AI systems need to learn to ask good questions, so they can acquire knowledge or help diagnose problems.
- The authors make a good effort in running Human evaluation experiments and present strong results.
- The paper is generally well written.

Weaknesses
- The paper is already out of date - ChatGPT-4V would presumably react quite differently and you could maybe run the entire study with GPT-4V, or now Bard (Gemini) -- how do results look like here?
- The paper does not go technically deep. Prompts have been manually generated, with ChatGPT in mind, and it is not clear if there was an attempt to optimize prompts for the other LLMs that were tested, or if prompts were simply rerun on different LLMs.
-

---

> ### Author Response · Authors · 2024-02-05
>
> Thank you for your feedback! Here we answer your questions below
>
> **Q1: Discuss the impact of recently released MM-LLMs. How are the results when combined with GPT-4V?**
>
>
> This paper's primary contribution is an automated questioning-answering conversation system that extracts more image information than a single model alone. Our contribution is perpendicular to the model designs. With more powerful MM-LLMs than BLIP-2 currently available, we conducted a new experiment utilizing GPT-4V, the strongest MM-LLM. In this experiment, GPT-4V functions as both questioner and answerer within the ChatCaptioner framework, while maintaining all other settings. Human evaluators were asked to choose the caption that accurately encompasses more image details, comparing GPT-4V's standalone generation to its application within the ChatCaptioner framework. Results demonstrate that in 97% of cases, captions from GPT-4V+ChatCaptioner provide more image information, validating the efficacy of our proposed framework in extracting further details through questioning and answering.
>
>
>
> | Method         | GPT-4V  | GPT-4V+ChatCaptioner |
> |----------------|---------|----------------------|
> | Human Votes    | 3/100   | 97/100                |
>
>
>
> **Q2: How were the prompts designed?  if there was an attempt to optimize prompts for the other LLMs that were tested**
>
>
> In our initial investigation, we observed that for advanced LLMs like ChatGPT, which possess strong instruction-following capabilities, generating informative questions is relatively simple through basic prompt design. Consequently, we didn't engage in extensive prompt searching or engineering within our framework. This also highlights our framework's advantage: it can be easily replicated using straightforward prompts without the need for cumbersome prompt engineering when employing cutting-edge LLMs. For earlier LLMs, such as FlanT5 or OPT, which predate ChatGPT, we did not notice strong question-asking abilities regardless of the prompts used during our preliminary exploration. Hence, for a clear comparison, we employ the same prompt as ChatGPT.
>
>
>
> **Q3: Cite works of question generation in NLP (diversity, quality, etc)**
>
>
> Thanks for the suggestion. We have enriched the first paragraph of our related work section by including several works in this field. The updated part is highlighted in blue.

---

> > ### Comment · Reviewer_ch9K · 2024-02-28
> > **Thanks for the response**
> >
> > Thank you for running the additional GPT-4V experiment and adding it to the paper. This, as well as the other changes in response to reviewer comments, makes the paper significantly stronger and answers my questions.

---

### Review · Reviewer_G3WG · 2024-01-26

**Summary Of Contributions:**

This paper proposes a system called ChatCaptioner, which leverages a pretrained vision-language model (VLM), such as BLIP-2, and a strong text-only LLM backbone (gpt-3.5-turbo in this work). They use the LLM to generate questions over 10 dialogue rounds, and use BLIP-2 to answer each generated question. Finally, they generate a dense caption by prompting the LLM to summarize the information from the dialogue in a few sentences. This system allows them to generate more information visual descriptions compared to BLIP-2 (which usually generates shorter captions).

**Audience:**

Yes

**Broader Impact Concerns:**

There are no major issues with the ethical implications of the work. It combines off-the-shelf LLM and VLMs, and is focused on image captioning which is not very high risk. However, it would be useful to describe how the human evaluators were sourced and compensated (i.e., do they meet local standards for minimum wage).

**Claims And Evidence:**

Yes

**Requested Changes:**

## Would strengthen the work:
- The authors report that GPT-3.5 was the best API-based model at the time of the paper. However, I’m curious if we can perform an ablation with the GPT-4 backbone. It would be great to compare the performance of GPT-4V (which can also do dense captioning) and ChatCaptioner (BLIP-2 + GPT-4 text-only). Does the proposed approach come close to the multimodal GPT-4? Even if not, ChatCaptioner seems like it would have significantly lower cost, so it may be an argument to use it (BLIP-2 + GPT-3.5) over the more expensive multimodal APIs (GPT-4V).
- How sensitive is the system to the number of questions? Does it saturate at 10 questions, or before that? Can we see an ablation experiment with different rounds of questioning? (maybe at least 3, 7, rounds, and 12 if it doesn’t saturate at 10)
- Would the automated caption metrics on Localized Narratives work better, since those captions are more detailed and presumably closer in distribution to the dense captions that ChatCaptioner generates?
- Instead of generating questions from an LLM, can we use the questions from the VisDial dataset to achieve similar results? What would be the difference in performance?

**Strengths And Weaknesses:**

Overall I think that the findings in the paper are quite interesting. I think the paper would be a lot stronger if there was more analysis to identify the degree of importance of each component in the system (prompt sensitivity, LLM, captioning model).

## Strengths:
- The paper proposes a simple method of leveraging vision-language models and strong LLMs to create a dense captioning system.
- They show that they can use the ChatCaptioner system to identify a greater coverage of image objects in an image (Sec 4.1), suggesting that this method is better at extracting information from BLIP-2, which is an interesting finding. It’s reminiscent of several automatic prompting approaches, which might be useful to discuss in related work.


## Weaknesses:
- The authors tested several other LLM backbones in 4.3, but only appear to report these results qualitatively (describing that only InstructGPT can generate meaningful questions). Could these results be quantitatively measured? For example, running human evaluations on a small subset of these to see the effects of different LLM backbones.
- Similar to the experiment in 4.3, can we do the same with other VLMs? It would be valuable to measure the performance of other captioning models, e.g., BLIP-1, MAGMA, Fromage, or LLaVA. It would be ideal to have some sort of chart showing the impact of different VLMs and LLMs: is one or the other more important?
- Missing references: VisDial [1] is highly relevant to this work. [2] on using LLMs for automatic prompt engineering seems highly relevant to the approach. It would be useful to discuss these papers (and corresponding follow up works) in the related work section.
- Minor typos: “BLLIP-2” in abstract, OpenAI API citation “(cha, 2023)” in Section 4


**References**

[1] Das, Abhishek, et al. "Visual dialog." Proceedings of the IEEE conference on computer vision and pattern recognition. 2017.

[2] Zhou, Yongchao, et al. "Large language models are human-level prompt engineers." arXiv preprint arXiv:2211.01910 (2022).

---

> ### Author Response · Authors · 2024-02-06
> **Official Comment by Authors Part I**
>
> Thank you for your feedback! Here we answer your questions below
>
>
> **Q1 The authors tested several other LLM backbones in 4.3, but only appear to report these results qualitatively (describing that only InstructGPT can generate meaningful questions). Could these results be quantitatively measured?**
>
> In Table 5 of the main paper, you can find the quantitative results showcasing the questioning ability of these LLM backbones. The assessment is based on the number of uniquely generated questions, providing insights into the tested LLMs' proficiency in generating diverse queries. The findings reveal that LLMs predating ChatGPT, such as OPT and FlanT5, exhibit relatively limited questioning abilities. On average, only about two out of the nine generated questions are distinct. In contrast, both InstructGPT and ChatGPT demonstrate a higher proficiency, rarely repeating questions within the same context.
>
>
> **Q2 It would be great to compare the performance of GPT-4V (which can also do dense captioning) and ChatCaptioner (BLIP-2 + GPT-4 text-only).**
>
>
>
> Here, we compare the caption quality between ChatCaptioner and GPT-4V using 20 COCO samples. GPT-4V's captions are preferred over ChatCaptioner's in all 20 examples judged by human annotators. Also, in 4 out of the 20 examples, BLIP-2 provides inaccurate answers. An example comparison using the image (http://images.cocodataset.org/val2017/000000117425.jpg) is shown below. Generally, GPT-4V's captions contain more details than ChatCaptioner's. Additionally, GPT-4V can provide more information about the object locations.
>
> ```
> GPT-4V:
> The image shows a young child staring at a small lit birthday candle on top of what appears to be a muffin or a small cake. The child is holding a fork and is standing close to a wooden table with the cake placed on a white plate. There's a mug to the left side of the plate, and you can see part of another person to the right side of the photo, though they are mostly out of frame. The scene suggests a birthday celebration, and the child seems curious or thoughtful about the lit candle.
>
> ChatCaptioner:
> In the image, a baby is standing in front of a wooden table. The baby is wearing a brown hoodie and appears to be curious. On the table, there is a small brown cupcake with a white candle. The baby is holding a fork and the cupcake, but is not looking directly at the camera.
> ```
>
>
>
>
>
> Additionally, we would like to highlight that the ChatCaptioner framework is compatible with other models like GPT-4V to enhance the captions further. Here, we conducted a new experiment where GPT-4V functions as both questioner and answerer in the ChatCaptioner framework. Human evaluators were asked to choose the caption that accurately encompasses more image details, comparing GPT-4V's standalone generation to its application within the ChatCaptioner framework. Results demonstrate that in 97% of cases, captions from GPT-4V+ChatCaptioner provide more image information, validating the efficacy of our proposed framework in extracting further details through questioning and answering.
>
>
>
> | Method         | GPT-4V  | GPT-4V+ChatCaptioner |
> |----------------|---------|----------------------|
> | Human Votes    | 3/100   | 97/100     |
>
>
>
> **Q3 Can we do an ablation study to replace the answerer with other VLMs?**
>
> Here, we conducted a new experiment where we replaced BLIP-2 with GPT-4V as the answerer while keeping the questioner the same. Experimental results below show that in 92% of the cases, the combination of ChatGPT questioner and GPT-4V answerer is preferred by human annotators over the original combination of ChatGPT questioner and BLIP-2 answerer, suggesting the importance of having a good answerer in our system. This may be caused by the incorrect answers from BLIP-2, as analyzed in Q2 and also in Table 3 of the main paper.
>
>
> |Questioner |  Answerer |  Human Votes  |
> |---------|-------------|-------------|
> | ChatGPT  | GPT-4V    |  92%|
> | ChatGPT  | BLIP-2   |  8%|
>
>
> **Q4 How sensitive is the system to the number of questions?**
>
>
> We set the predetermined number of Q&A rounds to 10. While theoretically, ChatGPT could acquire more information by posing additional questions, its questioning ability is not boundless. To delve into this, we scrutinized the queries generated across 30 Q&A rounds for 50 randomly selected COCO images. Notably, we observed instances where ChatGPT halted question generation after a certain number of rounds. On average, ChatGPT successfully engages in an effective dialogue for approximately 16.34 Q&A rounds, with a worst-case scenario of 12 rounds. These findings indicate that limiting the Q&A rounds to 10 is a prudent approach to circumvent ChatGPT's questioning limitations.

---

> > ### Author Response · Authors · 2024-02-06
> > **Official Comment by Authors Part II**
> >
> > **Q5 Would the automated caption metrics on Localized Narratives work better, since those captions are more detailed and presumably closer in distribution to the dense captions that ChatCaptioner generates?**
> >
> >
> > It wouldn't suffice. The automatic caption metric gauges the similarity between two sentences, but language's inherent flexibility allows for dissimilar sentences to convey the same content, and vice versa. This phenomenon becomes particularly pronounced with longer sentences. Consider this example: we randomly select a ground truth (GT) caption from Localized Narratives. A is a caption rephrased by ChatGPT, maintaining the same meaning. B, on the other hand, is a manipulated caption following the GT format but conveying a different meaning. The automated metric Rouge-L utilized by Localized Narratives indicates that B attains a significantly higher score than A, even though C portrays an entirely distinct scene. Consequently, automatic caption metrics on Localized Narratives face challenges in achieving accurate assessments.
> >
> > ```
> > GT caption: This picture is clicked outside on lawn,there is a wooden sofa with cushion pillow and in front there is table with flower vase,bowl and glass and above its a tree..
> >
> > A: Beneath the outstretched branches of a tree, a table adorned with a flower vase, a bowl, and a glass is prominently placed. Behind it lies a wooden sofa, equipped with a cozy cushioned pillow, positioned on a wide expanse of lawn.
> >
> > B: This picture is clicked outside on lake,there is a wooden boat with paddles and in the boat there is table with flower vase, bowl and glass and above its a tree.
> > ```
> >
> > *Rouge-L(GT, A) = 0.22*
> >
> > *Rouge-L(GT, B) = 0.84*
> >
> >
> >
> >
> >
> >
> > Furthermore, to underscore ChatCaptioner's efficacy in the context of Localized Narratives, we conducted a comparative analysis between ChatCaptioner's generated captions and the ground truth captions from Localized Narratives. The results reveal that ChatCaptioner receives twice the number of votes compared to the GT captions. This outcome suggests that our approach consistently provides richer image information, as judged by human evaluators.
> >
> >
> >
> >
> > | Method         |ChatCaptioner |  BLIP-2 |  GT  |
> > |----------------|---------|-------|-------------|
> > | Human Votes    | 60%   | 6.5%         |  33.5%|
> >
> >
> > **Q6 Typos and related works**
> >
> > Thank you for the suggestions. We have updated the related work section in the main paper and corrected the typos. The updated parts are highlighted in blue.

---

> > > ### Comment · Reviewer_G3WG · 2024-02-06
> > > **Thanks for the response**
> > >
> > > Thank you for running these additional experiments. I think that the paper would be strengthened with the inclusion of these results, especially the ablation experiments with GPT-4V and other captioning backbones. I think that it would still be great to have some automated metric since running human evaluations is often prohibitively expensive during development (maybe using [LLMs as a judge](https://arxiv.org/abs/2306.05685)?), but this is a minor concern.

---

> > > > ### Author Response · Authors · 2024-02-09
> > > >
> > > > Thank you for the suggestion. We have revised the paper to incorporate the GPT-4V-related experiments highlighted in blue in Section 4.4.

---

### Decision · Action_Editor_vYtA · 2024-03-04

**Recommendation:** Accept as is

**Comment:**

The reviewers were fairly aligned in their views of the paper, and I agree with them about it. The central idea of the paper, that of asking questions with an LLM, is an interesting one. However, the implementation in this work is a bit mundane, stapling existing prompted models together. As one reviewers pointed out in the private comments, the introduction of GPT-4V steals a little of this work's thunder, although the fact that asking questions still leads to higher performance is interesting. The addition of GPT-4V exploration to the work during the discussion period is very nice.

The evaluation is fairly thorough and sound, in my judgment. One reviewer pointed out that automatic evaluation could help future researchers evaluate on this task more consistently and cheaply.

Minor: I spotted one typo going through the paper. Section 4.4: CapCaptioenr <- typo in the section heading. Please fix for the final version.

**Audience:**

Yes, I believe so

**Claims And Evidence:**

Yes, the paper presents an image captioning system and uses a combination of human and automated analysis to support the main empirical claims about that system's performance. G3WG suggested some additional baselines and ch9K specifically mentions GPT-4V, which is now added to the paper.